# Synthesis and Micellization Behavior of Amphiphilic Block Copolymers of Poly(N-vinyl Pyrrolidone) and Poly(Benzyl Methacrylate): Block versus Statistical Copolymers

**DOI:** 10.3390/polym15092225

**Published:** 2023-05-08

**Authors:** Nikoletta Roka, Marinos Pitsikalis

**Affiliations:** Industrial Chemistry Laboratory, Department of Chemistry, National and Kapodistrian University of Athens, Panepistimiopolis Zografou, 15771 Athens, Greece; nikolettaroka@yahoo.gr

**Keywords:** poly(N-vinyl pyrrolidone) (NVP), poly(benzyl methacrylate) (BzMA), block copolymers, statistical copolymers, micelles, encapsulation of drugs

## Abstract

Block copolymers of N-vinyl pyrrolidone (NVP) and benzyl methacrylate (BzMA), PNVP-*b*-PBzMA, were prepared by RAFT polymerization techniques and sequential addition of monomers. The copolymers were characterized by Size Exclusion Chromatography (SEC) and NMR spectroscopy. Differential Scanning Calorimetry (DSC), Thermogravimetric Analysis (TGA) and Differential Thermogravimetry (DTG) were employed to study the thermal properties of these copolymers. The micellization behavior in THF, which is a selective solvent for the PBzMA blocks, was examined. For comparison the self-assembly properties of the corresponding statistical copolymers, PNVP-*stat*-PBzMA, were studied. In addition, the association behavior in aqueous solutions was analyzed for the block copolymers, PNVP-*b*-PBzMA. In this case, the solvent is selective for the PNVP blocks. Dilute solution viscometry, static (SLS) and dynamic light scattering (DLS) were employed as the tools to investigate the micellar assemblies. The efficient encapsulation of the hydrophobic curcumin within the micellar core of the supramolecular structures in aqueous solutions was demonstrated by UV-Vis spectroscopy and DLS measurements.

## 1. Introduction

The most important feature of block copolymers is their ability to self-assemble either in bulk or in selective solvents giving rise to extremely interesting supramolecular structures with unique properties [1,2,3,4]. Numerous studies have been devoted to the microphase separation phenomena in bulk [5,6] and to the micellization behavior in selective solvents [7,8,9]. Τhe subclass of amphiphilic block copolymers is ever more interesting, due to the possibility of forming micellar aggregates both in organic solvents and in aqueous solutions [10,11,12,13,14]. Therefore, these systems have found numerous and diverse applications including colloidal stabilization [12,15], latex technology [16], compatibilization of polymer blends [17], catalytic supports [18] and lithographic templates [19] in organic solvents, as well as controlled drug delivery [20,21,22,23], water purification [24,25], viscosity and surface modification [26,27,28,29] and nanoreactors [30,31] in aqueous solvents.

Recent advances in polymer synthesis have allowed for the preparation of a large variety of amphiphilic block copolymers. Controlled/living polymerization techniques, such as anionic, cationic, Atom Transfer Radical (ATRP), Nitroxide-Mediated Radical (NMRP), Reversible Addition Fragmentation Chain Transfer (RAFT), Group Transfer (GTP), Ring Opening (ROP) and coordination polymerization have been successfully employed towards this direction [32,33,34,35,36,37,38,39,40,41,42,43,44,45,46,47,48,49,50]. The most commonly used water soluble polymer is undoubtably poly(ethylene oxide) (PEO). It is nonionic, semicrystalline, biocompatible and biodegradable and commercially available in various molecular weights and with different end groups. Therefore, it has been very frequently employed in various biomedical applications [51,52,53,54,55]. However, PEO shows lower critical solution temperature (LCST), meaning that its solubility in aqueous solutions decreases upon increasing the temperature.

This problem can be resolved by replacing PEO with another water-soluble polymer having similar properties, with the exception of LCST. A suitable choice is poly(N-vinyl pyrrolidone) (PNVP), an amorphous polymer soluble in both aqueous solutions and in organic solvents, without LCST [56,57,58,59]. The main problem with this polymer is the lack of a suitable polymerization technique to provide control over the molecular characteristics and the ability to obtain complex macromolecular architectures. This obstacle was overcome by applying RAFT polymerization methodologies [60].

NVP can be polymerized exclusively via radical polymerization, because its amide keto group cannot be conjugated with the vinyl group. Conventional radical polymerization was employed for many years as the only source of polymers based on NVP [61,62]. However, under these experimental conditions, there is no control over the molecular characteristics and no possibility to obtain complex macromolecular architectures in a controlled way. Among the various controlled radical polymerization techniques, Atom Transfer (ATRP) and Nitroxide Mediated (NMP) Radical Polymerization have been shown to be inefficient methodologies to polymerize NVP [63]. On the contrary, RAFT has been proven to be the only available approach to provide control over the molecular weight and the dispersity of PNVP and the possibility to synthesize complex architectures based on PNVP [60]. In addition, RAFT can be combined with other controlled/living polymerization techniques, thus opening new horizons towards the synthesis of structures with unique properties and interesting applications.

The monomers employed in RAFT polymerization are classified in two categories, the more activated (MAMs) and the less activated monomers (LAMs), depending on their ability to stabilize radicals [64]. NVP is considered to belong to the family of LAMs. These monomers have a double bond connected to a saturated carbon atom or is conjugated to a lone pair on oxygen or nitrogen. Polymerization of these monomers produces poorly stabilized radicals. On the other hand, methacrylates, such as BzMA belong to the family of MAMs. For these monomers, the double bond is conjugated to an unsaturated system, such as nitrile, aromatic ring or carbonyl groups (e.g., the methacrylate monomers). Highly stabilized radicals are produced form these monomers, due to extended resonance effects. Consequently, the difference in reactivity between NVP and BzMA is considerably high, making it difficult to combine these monomers in a single polymerization system and thus rendering the synthesis of the respective block copolymers challenging.

In this work, block copolymers of N-vinyl pyrrolidone (NVP) and benzyl methacrylate (BzMA) were prepared by RAFT approaches and sequential addition of monomers following synthetic protocols given in Figure 1. In the past, statistical copolymers of these monomers (PNVP-*stat*-PBzMA) have been prepared and characterized [65]. Their thermal properties and especially the kinetics of their thermal decomposition have been thoroughly studied. PBzMA is hydrophobic and amorphous with a rather low glass transition temperature, *T*g _PBzMA_ = 54 °C, which is much lower than that of PSt, *T*g _PSt_ = 100 °C, implying the formation of more soft and easily processable copolymeric structures [66]. PBzMA has found many potential applications such as contact lenses, disinfectant hands gel [67], polymer optical fiber [68], monoliths for capillary electrochromatography [69], lithography [70], adhesives and coating [71].

The micellization behavior of the PNVP-*b*-PBzMA block copolymers in tetrahydrofuran (THF) in aqueous solutions is examined in this work. For comparison, the micellization behavior of the corresponding statistical copolymers is also examined. In this case, the self-assembly behavior was examined in THF, a selective solvent for the PBzMA sequences. The encapsulation ability of the hydrophobic compound curcumin within the micellar core was also tested in aqueous solutions of the block copolymers.

## 2. Materials and Methods

### 2.1. Materials

N-Vinyl pyrrolidone (≥97% FLUCA) containing sodium hydroxide as inhibitor was dried overnight over calcium hydride and was distilled prior to use. Benzyl methacrylate (TCI Chemicals, Chennai, India) stabilized with methyl hydroquinone, was also dried over calcium hydride overnight and then was distilled under vacuum prior the polymerization. Azobisisobutyronitrile AIBN (98% ACROS, Geel, Belgium) was purified by recrystallization twice from methanol and was then dried under vacuum. The Chain Transfer Agents, [(O-ethylxanthyl)methyl]benzene, CTA1, and O-ethyl S-(phthalimidylmethyl) xanthate, CTA2 were synthesized following literature protocols [72,73]. The synthesis of [(O-ethylxanthyl)methyl]benzene was conducted by stirring ethanol and KOH until a clear solution was formed. CS_2_ was added into the solution slowly, and the mixture was stirred for 10 h at room temperature before excessive CS_2_ was distilled off at 70 °C. Benzyl chloride in ethanol was added to afford the desired product. On the other hand, O-Ethyl S-(Phthalimidylmethyl) Xanthate was prepared from O-Ethyl xanthic acid potassium salt and N-(bromomethyl)phthalimide.

All other reagents and solvents were of commercial grade and were used as received.

### 2.2. Synthesis of PNVP-stat-PBzMA Statistical and PNVP-b-PBzMA Block Copolymers via RAFT Polymerization

The synthesis of the PNVP-*stat*-PBzMA statistical copolymers was accomplished in bulk at 60 °C employing AIBN as the initiator and [1-(O-ethylxanthyl)ethyl] benzene as the CTA, along with other CTAs. Additional details regarding the synthesis and characterization of these materials were given in a previous publication [65].

The block copolymers were prepared in glass reactors employing high vacuum techniques [74,75], O–ethyl S–(phthalimidymethyl) xanthate as the CTA and by sequential addition of monomers, starting from the polymerization of NVP. A typical polymerization procedure for NVP with final M_n_ = 10 × 10^3^ (Table 3, sample #4) with a molar ratio of [NVP]_0_/[CTA2]_0_/[AIBN]_0_ = 100/1/0.2 is described as follows: 5 g of NVP were polymerized in the presence of 0.1284 g CTA and 0.0148 g AIBN in 5 mL of benzene. The polymerization mixture underwent three freeze-thaw pump cycles in order to eliminate the oxygen from the polymerization flask. The reactor was flame-sealed and placed in a preheated oil-bath at 60 °C for 12 h.

The reaction was stopped by removing it from the oil-bath and cooling the mixture under the flow of cold water. The reactor was then opened so as to expose the mixtures to air. The polymer was precipitated in an excess of diethyl ether. This procedure was repeated three times in order to ensure the removal of any unreacted monomer residues. The polymers were subsequently dried overnight in a vacuum oven at 50 °C to remove any residual solvent.

The synthesis of the other PNVP homopolymers was conducted under the same experimental conditions, however, using the molar ratio of [NVP]_0_/[CTA2]_0_/[AIBN]_0_ = 300/1/0.2. The conversions for all homopolymers were near quantitative.

The block copolymerization reaction was performed in dioxane solutions at 80 °C for 3 days in glass reactors under high vacuum conditions. Quantities of the PNVP homopolymer, serving as the macro-CTA, the BzMA monomer, the AIBN radical initiator and the dioxane solvent are reported in Table 1. The polymerization mixture was subjected to three freeze-thaw pump cycles in order to eliminate the oxygen from the polymerization flask The polymerization was terminated by removing the reactor from the oil-bath and cooling the mixture under a flow of cold water. The reactor was then opened so as to expose the mixture to air. The polymer was precipitated in an excess of methanol. The crude product was dissolved in THF and reprecipitated in methanol. This procedure was repeated three times in order to ensure the removal of any unreacted monomer residues. Afterwards, the polymers were dried overnight in a vacuum oven at 50 °C to remove any residual solvent. The block copolymer #4 with the lowest PNVP content was further purified from the excess PNVP block that remained in the final product by dissolution of the excess PNVP block in ultrapure water and then removing the aqueous phase. The afforded polymer, insoluble in water, was dissolved in benzene and then precipitated in methanol.

### 2.3. Encapsulation Process

For the encapsulation process separate solutions were prepared in THF, one for the block copolymer and one for the hydrophobic drug curcumin. For example, for the sample PNVP-*b*-PBzMA#3 the concentration was equal to 1.78 × 10^−4^ g/mL. The solutions were left to stand overnight to achieve complete dissolution. The following day, the diblock copolymer solution was split into four portions, and a different amount of the curcumin solution was added to each one. After efficient mixing, the appropriate amount of extra pure water was added to the respective vials in order to have a final volume of 5 mL of solutions. Finally, THF was allowed to evaporate for several hours by heating at 65 °C. The final concentrations of curcumin varied for sample PNVP-*b*-PBzMA#3 from 2.89 × 10^−6^ g/mL to 1.08 × 10^−5^ g/mL. The same procedure was repeated for all the block copolymers.

### 2.4. Characterization Techniques

The molecular weight (M_w_) as well as the molecular weight distribution, Ð= M_w_/M_n_, were determined by size exclusion chromatography, SEC, employing a modular instrument consisting of a Waters model 510 pump, U6K sample injector, 401 differential refractometer and a set of 5μ-Styragel columns with a continuous porosity range from 500 to 10^6^ Å. The carrier solvent was CHCl_3_ and the flow rate 1 mL/min. The system was calibrated using nine Polystyrene standards with molecular weights in the range of 970–600,000.

The composition of the copolymers was determined from their ^1^H NMR spectra, which were recorded in chloroform-d at 30 °C with a 400 MHz Bruker Avance Neo spectrometer (Billerica, MA, USA).

For UV/Vis measurements, a Perkin Elmer Lamda 650 spectrophotometer (Waltham, MA, USA) was used from 250 to 800 nm, at room temperature, using a quartz cell of 3 mL.

The T_g_ values of the copolymers were determined by a 2910 Modulated DSC Model from TA Instruments (New Castle, DE, USA). The samples were heated under nitrogen atmosphere at a rate of 10 °C/min from −30 °C up to 220 °C. The second heating results were obtained in all cases.

The thermal stability of the copolymers was studied by thermogravimetric analysis (TGA) employing a Q50 TGA model from TA Instruments (New Castle, DE, USA). The samples were placed in a platinum pan and heated from ambient temperatures to 600 °C in a 60 mL/min flow of nitrogen at heating rates of 3, 5, 7, 10, 15 and 20 °C/min.

Refractive index increments, dn/dc, at 25 °C were measured with a Chromatix KMX-16 (Milton Roy, LDC Division, Riviera Beach, FL, USA) refractometer operating at 633 nm and calibrated with aqueous NaCl solutions.

Dynamic Light Scattering (DLS) measurements were conducted with a Brookhaven Instruments (Holtsville, NY, USA) Bl-200SM Research Goniometer System (Holtsville, NY, USA) operating at λ = 640 nm and with 40 mW power. Correlation functions were analyzed by the cumulant method and Contin software (Holtsville, NY, USA) [76]. The correlation function was collected at 45, 90, and 135°, at 25 °C.

The angular dependence of the ratio Γ/q^2^, where Γ is the decay rate of the correlation function and q is the scattering vector, was not very important for most of the micellar solutions. In these cases, apparent translational diffusion coefficients at zero concentration, D_o,app_, were measured using Equation (1):D_app_ = D_0, app_(1 + kDc) (1)
where k_D_ is the coefficient of the concentration dependence of the diffusion coefficient. Apparent hydrodynamic radii at infinite dilutions, R_h_, were calculated with the aid of the Stokes–Einstein Equation (2):R_h_ = kT/6πη_s_D_0_, app(2)
where k is the Boltzmann’s constant, T is the absolute temperature and η_s_ is the viscosity of the solvent.

Viscometric data were analyzed using the Huggins Equation (3):η_sp_/c = [η] + k_H_[η]^2^c + … (3)
and the Kraemer Equation (4):lnη_r_/c = [η] − k_K_[η]^2^c + … (4)
where η_r_, η_sp_ and [η] are the relative, specific and intrinsic viscosities respectively, k_H_ and k_K_ the Huggins and Kraemer constants, respectively. All the measurements were carried out at 25 °C. Cannon-Ubbelohde dilution viscometers equipped with a Schott-Geräte AVS 410 automatic flow timer were used. Viscometric radii, R_v_, were calculated from the Equation (5):R_v_ = (3/10πN_A_)^1/3^([η] M_w,app_)^1/3^
(5)
where M_w,app_ is the weight average molecular weight determined by light scattering measurements.

## 3. Results and Discussion

### 3.1. Synthesis of the Statistical and Block Copolymers of NVP and BzMA

The synthesis of the statistical copolymers PNVP-*stat*-PBzMA was described earlier [65]. Briefly. it was conducted in bulk at 60 °C, using AIBN as the initiator and [(O-ethylxanthyl)methyl]benzene as the CTA. The samples are denoted by two numbers indicating the molar feed ratios employed for the synthesis of these materials. The molecular characteristics of the samples are given in Table 2. 

The synthesis of block copolymers via RAFT between either LAMs or MAMs is a relatively easy task, since both monomers have similar reactivities and therefore, sequential addition of monomers promotes well-defined block copolymers [77,78,79,80]. However, the case of block copolymers between MAMs and LAMs is more complicated. Various approaches have been developed to promote the synthesis of fine products between monomers having different reactivities. Among them, the employment of universal [80,81,82,83] and switchable CTAs [84,85,86,87] has been reported. Universal CTAs is a special class of RAFT agents, which is able to promote the polymerization of both MAMs and LAMs, whereas switchable CTAs may lead to the same results after simple chemical transformation, such as protonation in acidic environment.

In the framework of the universal CTAS, O-ethyl S-(phthalimidylmethyl) xanthate (CTA 2) was employed. It is a well-known CTA providing very good control over the RAFT polymerization of LAMs. However, for MAMs it is not very efficient, providing poor control. It was employed for the controlled polymerization of NVP providing well-defined macromolecular CTAs (macro-CTAs), with relatively narrow molecular weight distribution and low molecular weight, very close to the stoichiometric values. Subsequent polymerization of BzMA, which belongs to the MAM family, was expected to be less controlled. In order to minimize the termination and other side reactions, and to achieve the best degree of control during the polymerization, the conversion of the polymerization of BzMA was not allowed to reach high values, i.e., less than 50% in all cases. Under these conditions, the PNVP macro-CTA efficiently initiated the polymerization of BzMA leading to the desired block copolymers. This was verified by the presence of single peaks in the SEC traces after the completion of the copolymerization reaction (Figure 1) and the NMR spectra (Figure 2), which confirmed the presence of both blocks in compositions close to stoichiometry values. However, the difficulties in the polymerization of BzMA were manifested by the broadening of the molecular weight distribution in the block copolymers, compared to the initial PNVP block (the results are shown in Table 3) and, in the case of sample #4 having the highest PBzMA content, by the presence of an excess of the PNVP macro-CTA that remained unreacted after the polymerization of BzMA. In this case the remaining amount of PNVP was eliminated by selective dissolution in aqueous solutions. Even with these limitations and drawbacks, this procedure was efficient enough to provide relatively well-defined PNVP-*b*-PBzMA block copolymers. More data are included in the Appendix A.

### 3.2. Thermal Properties

The thermal properties of the statistical copolymers have been examined and presented in detail previously [61]. The DSC data for the block copolymers are given in Table 4. Both components are amorphous characterized by a specific Tg value. The results for the homopolymers are the following: (Tg)_PBzMA_ = 54 °C and (Tg)_PNVP_ = 187.1 °C [65]. The first block copolymer with the highest PNVP component revealed two separate Tg transitions. These results lay within the respective values of the two homopolymers. This behavior indicates the presence of microphase separation in the sample. However, there is partial mixing in the system, meaning that the microphases of PNVP are “contaminated” with PBzMA chains, while the microphases of PBzMA are “contaminated” with PNVP chains. This partial mixing is promoted by the relatively low molecular weight of the two constituent blocks of the copolymeric structure. The other three copolymers have only one Tg value, which is considerably higher than that of the PBzMA homopolymer. These samples have a much higher composition in the polymethacrylate component. The short PNVP block is not able to promote the appearance of a second transition in the copolymer. However, the presence of these macromolecular chains leads to a substantial increase of the Tg value of the PBzMA block.

The thermal stability of the copolymers was examined by TGA and Differential Thermogravimetry, DTG, studies and was compared with the respective homopolymers. The results are given in Figure 3 and Figure 4 and in Table 5. PNVP presents a single decomposition peak having a maximum of 437.5 °C, showing that a rather simple decomposition mechanism takes place. On the other hand, PBzMA shows a different behavior. It is less thermally stable than PNVP, showing two different thermal degradation steps with maxima at 292.8 and 357.5 °C. It is possible that the first step can be attributed to the thermal removal of the side benzyl group followed by the decomposition of the main polymethacrylate chain. The block copolymers combine the behavior of their constituent blocks. Therefore, the range of decomposition temperatures is much wider in the case of the block copolymers, compared to the respective homopolymers. PNVP-b-PBzMA #1 with the highest PNVP content has a main decomposition peak in DTG close to that of the PNVP homopolymer. The contribution of the polymethacrylate block is manifested with the appearance of an extended shoulder at lower temperatures at the same range as the PBzMA homopolymer. The other samples resemble the thermal degradation behavior of the PBzMA homopolymer, since this is their major component. A clear second decomposition peak can be seen for Sample #2, attributed to the PNVP block, whereas samples #3 and #4 with very low NVP contents present only weak shoulders at higher temperatures.

### 3.3. Self-Assembly Behavior of the Statistical Copolymers in THF

The micellization behavior of the statistical copolymers was examined in THF, a selective solvent for PBzMA [88]. The same study in aqueous solutions, where PNVP is soluble, was not possible, since stable micellar solutions were not obtained due to the low PNVP content of the copolymers and the statistical distribution of the NVP monomer units across the copolymer chain. Static and dynamic light scattering along with dilute solution viscometry measurements were conducted in CHCl_3_, a common good solvent for both homopolymers PNVP and PBzMA and in the selective solvent THF as well. The results from these measurements are given in Table 6, Table 7 and Table 8 for both solvents. Characteristic plots from SLS, DLS and viscometry measurements are given in Figure 5, Figure 6 and Figure 7. Additional plots are provided in the Appendix A.

The static light scattering experiments on the statistical copolymers revealed that the degrees of association are very low in THF. This conclusion can be attributed to the rather low NVP content of the copolymers and, most importantly, to the statistical distribution of the NVP monomer units along the copolymer chain. It was found that the terminal model is applicable for this system, with much smaller reactivity ratios for NVP compared to those for BzMA. In addition, the NVP-NVP dyad sequence content is very low. Therefore, the NVP units are surrounded by much longer BzMA units, thus rendering their organization in stable extended micellar cores difficult. This rather weak self-assembly behaviour is expected and is common in the case of statistical copolymers. However, the low second virial coefficient, A_2_ values indicate the presence of strong intramolecular, rather than intermolecular interactions between the polymer chains.

These conclusions were further supported for the statistical copolymers by the DLS measurements. The plots D vs c were linear with small k_D_ values. This result is expected since k_D_ and A_2_ are related through the Equation (6):k_D_ = 2A_2_M + k_f_ − u (6)
where M is the molecular weight, k_f_ the coefficient of the concentration dependence of the friction coefficient and u the partial specific volume of the polymer. Due to the low degrees of aggregation in THF, low values of A_2_ imply low k_D_ values as well. Furthermore, CONTIN analysis revealed the presence of a single population of polymeric materials in THF. These structures are relatively polydisperse as evidenced by the second moment values μ_2_/Γ^2^ of the cumulant analysis, which were considerably higher than 0.1 for all samples. DLS measurements were also conducted in CHCl_3_, which is a common good solvent for both components, PNVP and PBzMA. The k_D_ values in CHCl_3_ were substantially higher than those measured in THF, since the latter solvent is a selective solvent, and the interactions with the statistical copolymers are less pronounced compared to the common good solvent. In addition, the R_h_ values in THF were found to be lower than those measured in CHCl_3_, indicating the formation of unimolecular micelles or small and compact micellar structures of very low degrees of association, as already evidenced by SLS measurements.

Dilute solution viscometry was also employed to further study the aggregation behavior of the statistical copolymers in THF solutions. The intrinsic viscosity values in THF solutions are lower than those measured in CHCl_3_. This is a direct indication of the existence of either unimolecular micelles or rather compact micelles of low degree of association in THF. These results further confirm the conclusions derived from the SLS and DLS measurements. The ratio of the viscometric over the hydrodynamic radius, R_v_/R_h_, from viscometric and DLS measurements respectively, is equal to unity in CHCl_3_ solutions, revealing the presence of spherical structures, as expected in a common good solvent. The R_v_ values in THF are slightly higher than those measured in CHCl_3_, thus showing only a small tendency for association in the selective solvent. The same result is concluded judging from the ratio R_v_/R_h_ in THF solutions, which varies between 0.94 to 1.08, indicating a less pronounced association process and the presence of unimolecular or small compact spherical micelles.

### 3.4. Self-Assembly Behavior of the Block Copolymers in THF and in Aqueous Solutions

The aggregation behavior of the block copolymers was examined in both THF and aqueous solutions using SLS and DLS techniques. A similar experimental behavior was also found in the case of the PBzMA-*b*-PNVP block copolymers in THF as in the case of the statistical copolymers. In this case it was expected that the degrees of association should be much higher, since the self-assembly process could be promoted by the block nature of the copolymers. However, the N_w_ values were not higher than the corresponding values found for the statistical copolymers, even for sample #1 with the highest content in NVP. This result indicates that THF does not promote the formation of multimolecular micelles. Nevertheless, the low A_2_ values from SLS measurements and the low k_D_ values from DLS measurements imply that self-assembly takes place in THF for the block copolymers, leading to the formation of unimolecular micelles or low aggregation number spherical micelles, as in the case of the corresponding statistical copolymers. The R_h_ values in THF were also lower or slightly higher than the corresponding R_h_ values measured in CHCl_3_, indicating the presence of small and compact micelles in solution. CONTIN analysis confirmed these conclusions, indicating the presence of single populations without any angular dependence and with rather lower μ_2_/Γ^2^ values, compared to the CHCl_3_ solutions. This behavior is attributed to the reduced interactions of the copolymers with the selective solvent and reflects the formation of unimolecular micelles or even small spherical aggregates. These results indicate that THF is not able to promote the formation of high aggregation number micelles. The data in THF solutions are summarized in Table 9 and Table 10, and characteristic plots from the SLS and DLS measurements are provided in Figure 8 and Figure 9. Additional data are included in the Appendix A.

The self-assembly behavior was also studied in aqueous solutions. In this case, the statistical copolymers were not soluble. The rather low PNVP content and the relatively random distribution of the NVP monomer units along the copolymer chain make the stabilization of supramolecular micellar structures extremely difficult. On the other hand, it was much easier to have stable micellar solutions from the block copolymers in aqueous solutions. In order to achieve equilibrium structures, solutions of the blocks were prepared in THF, and the desired amount of water was added gradually. The solutions were heated for several hours at 50 °C to facilitate complete dissolution and were then further heated at 60 °C in order to gradually evaporate THF. The procedure led to stable micellar solutions in aqueous environment, with the characteristic blue tint, indicating the presence of large aggregates.

SLS measurements confirmed the formation of micelles with very high aggregation numbers. The A_2_ values were very low, actually close to zero for all copolymers, as a result of the very weak interactions of the copolymers with the selective solvent. Judging from the composition of the copolymers, it can be concluded that crew-cut micelles are formed in aqueous solutions for samples #2, #3 and #4 [89,90], whereas star-like micelles for sample #1. [91] High Rg values were also obtained from the Zimm plots in aqueous solutions. Comparing these data with the extremely high micellar molecular weights, it can be concluded that the supramolecular structures are very compact with extended cores (Figure 10).

These conclusions were further confirmed by DLS measurements. A characteristic example is given in Figure 11. The k_d_ values were considerably higher than those measured in CHCl_3_ or in THF, obviously reflecting the tremendous increase of the molecular weight of the supramolecular structures, which are formed in aqueous solutions. CONTIN analysis revealed the presence of single populations of rather low dispersity (μ_2_/Γ^2^ values lower than 0.2 in almost all samples and all studied concentrations). In addition, no angular dependence was observed, and the aggregates were thermally stable up to 60 °C, without any indication of disassociation or further aggregation to higher supramolecular structures. Finally, the R_h_ values, were much higher than those measured in the common good solvent CHCl_3_ or in the selective solvent THF. Nevertheless, these values were not as high as expected for the huge molecular weights obtained by SLS measurements. The ratio R_g_/R_h_ was close to unity for all the copolymers. All these findings point to the conclusions that stable, spherical and compact micelles exist in aqueous solutions. The high content of the hydrophobic component and the strong intra- and intermolecular interaction of the aromatic rings of the polymethacrylate’s side groups seem to be responsible for the formation of stable and probably dry micellar cores. The high solubility of the PNVP corona forming chains in water ensures the solubilization of the final supramolecular structure. Additional data are included at the SIS (Appendix A).

### 3.5. Encapsulation of Curcumin into the Micellar Solutions

Several studies have appeared in the literature regarding the encapsulation of hydrophobic compounds, especially drugs, within the micellar core of amphiphilic copolymers containing PNVP as the water-soluble material [92]. In addition, drug delivery systems, based on PNVP, such as microparticles, nanoparticles, fibers, films, tablets and hydrogels have been employed in the past. In the present study, curcumin was chosen as the hydrophobic compound to test the encapsulation ability of block copolymers of PNVP with polymethacrylates.

Curcumin is a polyphenolic compound (Figure 2) exhibiting numerous interesting properties, including antioxidant, antibacterial, antimicrobial, antifungal, anti-inflammatory and anti-carcinogenic activities [93,94,95,96]. It has been used for the treatment of Alzheimer’s disease, multiple myeloma, psoriasis, myelodysplastic syndrome, and anti-human immunodeficiency virus cycle replication. Several efforts have been devoted to efficiently encapsulate curcumin in block copolymer micelles [97].

UV-vis spectroscopy and DLS measurements were conducted to study the encapsulation ability of curcumin within the block copolymer micelles. A calibration curve from the absorbance values at 423.50 nm of curcumin solutions in THF with concentration was recorded. The results are given in the SIS in Appendix A. Using this calibration curve and measuring the absorbance of the polymer solutions with the encapsulated curcumin, the drug loading capacity, DLC, and the drug loading efficiency, DLE, were calculated employing the following Equations (7) and (8):(7)$$\mathrm{drug}\;\mathrm{loading}\;\mathrm{capacity}\;\left(\mathrm{DLC}\right)\%=\frac{\mathrm{mass}\;\mathrm{of}\;\mathrm{loaded}\;\mathrm{drug}}{\mathrm{mass}\;\mathrm{of}\;\mathrm{polymer}}\times 100$$
(8)$$\mathrm{drug}\;\mathrm{loading}\;\mathrm{efficiency}\;\left(\mathrm{DLE}\right)\%=\frac{\mathrm{mass}\;\mathrm{of}\;\mathrm{loaded}\;\mathrm{drug}}{\mathrm{mass}\;\mathrm{of}\;\mathrm{drug}\;\mathrm{in}\;\mathrm{feed}}\times 100$$

Characteristic sectra from the encapsulation of various amounts of curcumin within the micellar core of the block copolymer PNVP-b-PBzMA #3 are given in Figure 12, whereas the complete data with the DLC and DLE values are given in Table 11. R_h_ values from DLS measurements for the polymer solution prior to and after the encapsulation of curcumin are also included in Table 11. Characteristic DLS CONTIN plots are given in Figure 13.

It is evident from these results that curcumin was efficiently entrapped in the core of the micellar solutions in aqueous solutions. The solutions were stable for several weeks and had the characteristic yellow color of curcumin. With targeted entrapment levels up to 7.5% the DLC was fairly high, and in addition the DLE reached high levels. Upon increasing the hydrophobic content of the block copolymers, the DLE values increased substantially. The PNVP-b-PBzMA #1 sample with the lowest composition in PBzMA showed the lowest DLE values, up to almost 40%, as revealed in Table 11. However, the other samples with much higher contents in PBzMA presented very high DLE values up to more than 90%. Compared to other similar copolymeric systems that have been employed for the encapsulation of curcumin, these results are very promising and confirm that the specific system has a very high ability to encapsulate hydrophobic compounds. The π-π interactions between the phenyl groups of curcumin and benzyl methacrylate may be responsible for the high DLE values.

DLS measurements were conducted for the initial micellar solutions and the same solutions after the encapsulation of the hydrophobic drug. In all cases, single populations were observed by CONTIN analysis. The R_h_ values from sample #1 were relatively stable before and after the entrapment of curcumin. This behavior can be attributed to the low PBzMA content. The micellar core is very small compared to the overall size of the nanoparticles. Therefore, the encapsulation of curcumin is not going to drastically affect the R_h_ value of the micelles. The opposite behavior was observed for the other block copolymers, where a substantial increase of the R_h_ values was obvious after the encapsulation of curcumin. These copolymers had a much higher PBzMA content, meaning that the overall micellar size is mainly affected by the size of the polymethacrylate core. Therefore, the entrapment of curcumin inside the micellar core leads to a pronounced increase in the overall R_h_ value of the nanoparticles.

## 4. Conclusions

Block copolymers of N-vinyl pyrrolidone (NVP) and benzyl methacrylate (BzMA), PNVP-*b*-PBzMA, were prepared by the RAFT methodology and sequential addition of monomers, starting from the polymerization of NVP, employing O–ethyl S–(phthalimidymethyl) xanthate was employed as the universal CTA. Rather well-defined products were obtained as was revealed by Size Exclusion Chromatography (SEC) and NMR spectroscopy. In the case of samples with very high polymethacrylate content the crude products were contaminated by PNVP homopolymers, which were effectively purified by extraction with water. Differential Scanning Calorimetry (DSC), Thermogravimetric Analysis (TGA) and Differential Thermogravimetry (DTG) were employed to study the thermal properties of these copolymers. Partial or complete mixing was observed depending on the composition and the molecular weights of the block copolymers. The thermal stability was influenced by both components in the case of the block copolymers. The micellization behavior in THF, which is a selective solvent for the PBzMA blocks, was examined. For comparison the self-assembly properties of the corresponding statistical copolymers, PNVP-*stat*-PBzMA, were studied. In addition, the association behavior in aqueous solutions was analyzed for the block copolymers, PNVP-*b*-PBzMA. In this case, the solvent is selective for the PNVP blocks. Dilute solution viscometry, static (SLS) and dynamic light scattering (DLS) were employed as the tools to investigate the micellar assemblies. It was found that THF is not able to promote the formation of high associates either in the case of the statistical or the block copolymers. Mainly unimolecular or low degree of micellization supramolecular structures were obtained in THF. On the other hand, stable, spherical micelles with very high degrees of association were observed in aqueous solutions. The efficient encapsulation of curcumin within the micellar core of the supramolecular structures was demonstrated by UV-Vis spectroscopy and DLS measurements.

## Data Availability

The data are available upon request.

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
