# Peer review of "Synthesis and Micellization Behavior of Amphiphilic Block Copolymers of Poly(N-vinyl Pyrrolidone) and Poly(Benzyl Methacrylate): Block versus Statistical Copolymers"

_polymers, 2023, doi:10.3390/polym15092225_

Round 1

Reviewer 1 Report

This manuscript presents a thorough study of amphiphilic block copolymers of poly(N-vinyl pyrrolidone)-b-poly(benzyl methacrylate). The statements are well-supported. The manuscript is easy to understand, however, I recommend minor editing of English. Overall, I recommend publication after minor revision. I have several questions and would like the authors to clarify. 

1. Are the statistical copolymers random or blocky? Is there any way to characterize it? If not, is there any way to do an estimation using their reactivity ratios?

2. Suggest summarizing the Rg/Rh values into some of the Tables. The sample names can be cleared up for easier understanding. 

3. What do the error bars look like? Statistical analysis is needed.

4. References are needed to justify "Judging from the composition of the copolymers, it can be concluded that crew-cut micelles are formed in aqueous solutions for samples #2, #3, and #4, whereas star-like micelles for sample #1".

5. Zimm plots need to be cleared up.

This manuscript presents a thorough study of amphiphilic block copolymers of poly(N-vinyl pyrrolidone)-b-poly(benzyl methacrylate). The statements are well-supported. The manuscript is easy to understand, however, I recommend minor editing of English. Overall, I recommend publication after minor revision. 

Author Response

Reviewer 1

We are grateful to the Reviewers for their fruitful comments. We took everything into consideration and we made the appropriate changes to our manuscript.

This manuscript presents a thorough study of amphiphilic block copolymers of poly(N-vinyl pyrrolidone)-b-poly(benzyl methacrylate). The statements are well-supported. The manuscript is easy to understand, however, I recommend minor editing of English. Overall, I recommend publication after minor revision. I have several questions and would like the authors to clarify. 

  1. Are the statistical copolymers random or blocky? Is there any way to characterize it? If not, is there any way to do an estimation using their reactivity ratios?

The synthesis of the statistical copolymers has been described in detail in a previous publication: Roka, N.; Pitsikalis M. Statistical Copolymers of N-Vinylpyrrolidone and Benzyl Methacrylate via RAFT: Monomer Reactivity Ratios, Thermal Properties and Kinetics of Thermal Decomposition.”  J. Macromol. Sci., Part A Pure Appl. Chem. 55, 222-230 (2018). The best way to characterize the nature of the statistical copolymers is by measuring the monomer reactivity ratios and the distribution of the monomer units along the copolymeric chains. As was expected, the reactivity ratio of benzyl methacrylate, BzMA, is much higher than that of N-vinyl pyrrolidone, NVP, meaning that we have a pseudo-diblock or a gradient copolymer. The final structure consists from PBzMA and PNVP end-blocks and an intermediate one with decreasing concentration of BzMA units and at the same time increasing NVP units.  

  1. Suggest summarizing the Rg/Rh values into some of the Tables. The sample names can be cleared up for easier understanding. 

The Rg/Rh values were incorporated in Table 10, as suggested by the Reviewer.

  1. What do the error bars look like? Statistical analysis is needed.

At least three measurements were taken for most of our experiments. It was found that the error for the statistical copolymers is around 5% (viscometry and scattering methods), whereas for the block copolymers it was up to 10%. The error bars were not given in our graphs to avoid the complexity for the reader. This is a rather typical behavior for the self-assembly of block copolymers.

  1. References are needed to justify "Judging from the composition of the copolymers, it can be concluded that crew-cut micelles are formed in aqueous solutions for samples #2, #3, and #4, whereas star-like micelles for sample #1".

We added the requested references to justify our statements, as the Reviewer suggested. It is well known, that when the micelles have a very extended core over the corona they are characterized as crew-cut micelles, whereas in the opposite situation (small core over the corona) the micelles as described as star-like structures.

  1. Zimm plots need to be cleared up.

 We tried to do our best. This is the outcome from the software of our instrument.

Comments on the Quality of English Language

This manuscript presents a thorough study of amphiphilic block copolymers of poly(N-vinyl pyrrolidone)-b-poly(benzyl methacrylate). The statements are well-supported. The manuscript is easy to understand, however, I recommend minor editing of English. Overall, I recommend publication after minor revision. 

The manuscript was edited by an English speaking person and minor mistakes were corrected.

Reviewer 2 Report

The manuscript by Roka et al. is an interesting study about a new Poly(N-vinyl Pyrrolidone) - Poly(Benzyl Methacrylate) blockcopolymer. The polymer synthesis is described very well, and the manuscript offers a lot of background on the polymers and their synthesis. However, the part including the analysis and especially the drug encapsulation lacks description and clear explanation of the data.

Here are some comments that would improve the manuscript, in my opinion:

1.       The dots in the title make it hard to read. Please shorten the title

2.       Page 2; “The Chain Transfer Agents, [(O-ethylxanthyl)me-thyl]benzene, CTA1, and O-ethyl S-(phthalimidylmethyl) xanthate, CTA2 were synthe-sized following literature protocols [68,69].”Please provide a brief description (about 2 sentences) of the synthesis.

3.       Page 3;” For the encapsulation process separate solutions were prepared in THF, one for the block copolymer and one for the hydrophobic drug curcumin” Which concentration of curcumin and of the polymer were used?

4.       Page 3; “the appropriate amount of extra pure water”: What’s the appropriate amount? Please give the volume of water added.

5.       Page 5: The first two paragraphs do not belong in the Results section but in the Introduction. In general, the Results part is very loaded with background knowledge. The content is very well written and interesting but should be concentrated and moved to the Introduction part.

6.       Page 6; “the broadening of the molecular weight distribution in the block copolymers,” please mention the table where we find these results.

7.       Page 6, “More data are included in the Supporting Information Section, SIS.”: Please describe the data and name the tables/figure, not just “in the SIS.”

8.       Page 8, Figure 2: Please show the integrals of all signals.

9.       Page 10, Table 6: What is scheme 103?

9.       Page 17, “The results are given in the SIS.”: Please describe the data and name the tables/figure, not just “in the SIS.”

9.        Page 17; “Using this calibration curve and measuring the absorbance of the polymer solutions with the encapsulated curcumin, the drug loading capacity, DLC, and the drug loading efficacy”: It is not fully clear to me how the authors distinguish between the concentration of the free drug and the concentration of the loaded drug using UV-Vis. Also, this description belongs to the Materials and Methods part.

9.      Page 17; “the block copolymer PNVP-b-PBzMA #1” The caption in the Figure 12 says #3.

9.    Page 17; “It is evident from these results that curcumin was efficiently entrapped in the core of the micellar solutions in aqueous solutions.” For me, it is not evident. Please explain in more detail what is seen in the figure and how you deduced the entrapment.

9.     Page 17; “The PNVP-b-PBzMA #1 sample with the lowest composition in PBzMA showed the lowest DLE values, up to almost 40%. “: Where are these data presented?

9.      Page 18; „The π-π interactions between the phenyl groups of curcumin and benzyl methacrylate may be responsible for the high DLE values.”: This sound like a correction of the data is needed.

9.       Page 19; Table 11: DLE of significantly above 100% seems odd. This sounds like there is a problem with the data.

The English language is acceptable. Nevertheless, some minor changes and some fine-tuning might be helpful.

Author Response

Reviewer 2

We are grateful to the Reviewers for their fruitful comments. We took everything into consideration and we made the appropriate changes to our manuscript.

The manuscript by Roka et al. is an interesting study about a new Poly(N-vinyl Pyrrolidone) - Poly(Benzyl Methacrylate) block copolymer. The polymer synthesis is described very well, and the manuscript offers a lot of background on the polymers and their synthesis. However, the part including the analysis and especially the drug encapsulation lacks description and clear explanation of the data.

Here are some comments that would improve the manuscript, in my opinion:

  1. The dots in the title make it hard to read. Please shorten the title

      The title has been modified according to the suggestion of the Reviewer to: “Synthesis and micellization behavior of amphiphilic block copolymers of poly(N-vinyl pyrrolidone) and poly(benzyl methacrylate). Block versus statistical copolymers”

  1. Page 2; “The Chain Transfer Agents, [(O-ethylxanthyl)methyl]benzene, CTA1, and O-ethyl S-(phthalimidylmethyl) xanthate, CTA2 were synthesized following literature protocols [68,69].”Please provide a brief description (about 2 sentences) of the synthesis.

      A brief description for the synthesis of the CTAs was added, as suggested by the Reviewer.

  1. Page 3;” For the encapsulation process separate solutions were prepared in THF, one for the block copolymer and one for the hydrophobic drug curcumin” Which concentration of curcumin and of the polymer were used?

      The appropriate details were added in the experimental part.

  1. Page 3; “the appropriate amount of extra pure water”: What’s the appropriate amount? Please give the volume of water added.

      We need a final volume for the solution of about 5 ml in order to conduct the UV-Vis and dynamic light scattering measurements. This detail was added to the text.

  1. Page 5: The first two paragraphs do not belong in the Results section but in the Introduction. In general, the Results part is very loaded with background knowledge. The content is very well written and interesting but should be concentrated and moved to the Introduction part.

      We thought that this knowledge is necessary to the broad scientific community to understand the difficulties regarding the synthesis of these block copolymers. That is why this part was included at the beginning of the Results and Discussion section. However, since the Reviewer proposed we moved this part to the Introduction section.

  1. Page 6; “the broadening of the molecular weight distribution in the block copolymers,” please mention the table where we find these results.

     These results refer to Table 3. We made the appropriate addition in the text.

  1. Page 6, “More data are included in the Supporting Information Section, SIS.”: Please describe the data and name the tables/figure, not just “in the SIS.”

     We added the suggested information in the text.

  1. Page 8, Figure 2: Please show the integrals of all signals.

      We added the integrals in the figure.

  1. Page 10, Table 6: What is scheme 103?

      We are sorry for the mistake. It is not “scheme 103”. The correct statement is “sample”

  1. Page 17, “The results are given in the SIS.”: Please describe the data and name the tables/figure, not just “in the SIS.”

     We added the suggested information in the text.

  1. Page 17; “Using this calibration curve and measuring the absorbance of the polymer solutions with the encapsulated curcumin, the drug loading capacity, DLC, and the drug loading efficacy”: It is not fully clear to me how the authors distinguish between the concentration of the free drug and the concentration of the loaded drug using UV-Vis. Also, this description belongs to the Materials and Methods part.

      Curcumin has negligible solubility in water. Practically, it is insoluble in water. If there was any turbidity in the solution or even a trace of insoluble curcumin the solutions were filtered prior taking the measurements. Therefore, the signal of the UV-Vis spectra reflect only the amount of encapsulated curcumin, since there is no free drug in the aqueous solution.

  1. Page 17; “the block copolymer PNVP-b-PBzMA #1” The caption in the Figure 12 says #3.

      We are sorry for the mistake. The Figure refers to sample PNVP-b-PBzMA #3.

  1. Page 17; “It is evident from these results that curcumin was efficiently entrapped in the core of the micellar solutions in aqueous solutions.” For me, it is not evident. Please explain in more detail what is seen in the figure and how you deduced the entrapment.

      As stated before, curcumin is almost insoluble in aqueous solutions. The characteristic and stable for several weeks yellow color of the micellar solutions (indicated in the graphical abstract of this manuscript) is a direct evidence for the efficient encapsulation of the drug into the micellar core. By the UV-Vis spectra we can measure the absorption of the micellar solutions at the wavelength of curcumin absorption. Then employing the calibration curve we can extract quantitative conclusions about the entrapped amount of the drug within the cores of the micelles. This is a typical procedure employed for hydrophobic compounds, including curcumin.  

  1. Page 17; “The PNVP-b-PBzMA #1 sample with the lowest composition in PBzMA showed the lowest DLE values, up to almost 40%. “: Where are these data presented?

   These data are included in Table 11. We made the appropriate addition in the text.

  1. Page 18; „The π-π interactions between the phenyl groups of curcumin and benzyl methacrylate may be responsible for the high DLE values.”: This sound like a correction of the data is needed.

     With this statement we tried to find a reasonable explanation about the relatively high DLE values for the entrapment of curcumin into our micellar structures. These π-π interactions may be responsible for this experimental evidence.

  1. Page 19; Table 11: DLE of significantly above 100% seems odd. This sounds like there is a problem with the data.

      We are sorry for this mistake. Of course, it is not reasonable DLE to exceed the 100% value. The correct results were added in the Table.

Comments on the Quality of English Language

The English language is acceptable. Nevertheless, some minor changes and some fine-tuning might be helpful.

The manuscript was edited by an English speaking person and minor mistakes were corrected.